# Molecular Management of High-Grade Serous Ovarian Carcinoma

**DOI:** 10.3390/ijms232213777

**Published:** 2022-11-09

**Authors:** Paula Punzón-Jiménez, Victor Lago, Santiago Domingo, Carlos Simón, Aymara Mas

**Affiliations:** 1Carlos Simon Foundation, INCLIVA Health Research Institute, 46010 Valencia, Spain; 2Department of Gynecologic Oncology, La Fe University and Polytechnic Hospital, 46026 Valencia, Spain; 3Department of Obstetrics and Gynecology, CEU Cardenal Herrera University, 46115 Valencia, Spain; 4Department of Pediatrics, Obstetrics and Gynecology, Universidad de Valencia, 46010 Valencia, Spain; 5Department of Pediatrics, Obstetrics and Gynecology, Beth Israel Deaconess Medical Center, Harvard University, Boston, MA 02215, USA; 6Department of Obstetrics and Gynecology, Baylor College of Medicine, Houston, TX 77030, USA

**Keywords:** ovarian cancer, high-grade serous ovarian carcinoma, early diagnosis, molecular biomarkers, liquid biopsy

## Abstract

High-grade serous ovarian carcinoma (HGSOC) represents the most common form of epithelial ovarian carcinoma. The absence of specific symptoms leads to late-stage diagnosis, making HGSOC one of the gynecological cancers with the worst prognosis. The cellular origin of HGSOC and the role of reproductive hormones, genetic traits (such as alterations in P53 and DNA-repair mechanisms), chromosomal instability, or dysregulation of crucial signaling pathways have been considered when evaluating prognosis and response to therapy in HGSOC patients. However, the detection of HGSOC is still based on traditional methods such as carbohydrate antigen 125 (CA125) detection and ultrasound, and the combined use of these methods has yet to support significant reductions in overall mortality rates. The current paradigm for HGSOC management has moved towards early diagnosis via the non-invasive detection of molecular markers through liquid biopsies. This review presents an integrated view of the relevant cellular and molecular aspects involved in the etiopathogenesis of HGSOC and brings together studies that consider new horizons for the possible early detection of this gynecological cancer.

## 1. Introduction

Ovarian cancer (OC), situated among the most aggressive and deadly gynecological malignancies, is the fifth leading cause of cancer-related death in females in developed countries, with a total of 313,959 new cases diagnosed in 2020 worldwide (66,693 in European countries) and 12,810 predicted annual deaths and 19,880 predicted diagnoses in 2022 in the US [1,2,3]. OC refers to a heterogeneous set of neoplasms subdivided according to genetics, histological evaluations, and tissue of origin [4,5,6]. The main subtypes include sex-cord stromal OC, germ-cell OC, and epithelial ovarian carcinoma (EOC) as the most common subtype, accounting for up to 95% of all cases [7,8]. EOC can be subdivided into five histological subtypes: mucinous, endometrioid, clear-cell, low-grade (LGSOC), and high-grade (HGSOC). HGSOC, the most common histological subtype, constitutes 70% of diagnosed EOC cases [9,10] and is often first diagnosed at advanced stages (III and IV), where the tumor has spread to the abdomen or outside the abdominal cavity [11,12], due to the lack of specific symptoms. A late diagnosis dramatically reduces therapeutic responses and overall survival rates in affected patients [13]; therefore, a strong, urgent rationale supports the design of early detection strategies, as tumor detection at stage I (i.e., confined to the ovaries) or II (i.e., confined to the pelvic area) could improve overall five-year survival rates [4,11,12,14]. On the other hand, the differential diagnosis of ovarian masses represents a challenge for clinicians [15]. A diagnosis can be made intraoperatively or weeks after surgery, which leads to a poor optimization of the hospital’s surgical resources and the need for re-interventions, with consequent associated costs to the health system [16,17].

Current clinical approaches that guide physicians in managing EOC remain primarily based on imaging, histological evaluation, serum markers such as CA125, or predictive models, which do not display sufficient sensitivity and sensibility [18,19]; however, a growing trend exists in exploring novel/alternative molecular tools [20]. In this review, we discuss the cellular origin of EOC and the role of hormones, genetic and epigenetic processes, and disrupted signaling pathways, and explore the diagnosis, prognosis, and response to therapy of HGSOC. Finally, we summarize current research aims regarding the identification of new biomarkers based on liquid biopsy and next-generation sequencing (NGS) to offer an integrated view of the progress made in the last decade toward the early and non-invasive diagnosis of EOC.

## 2. Current Molecular Approaches for the Diagnosis and Prognosis of HGSOC

Difficulties in the early detection of HGSOC, before the disease develops to advanced stages, can be attributed to the lack of specific symptoms, which are usually missed or attributed to other pathologies [21]. In clinical practice, the diagnosis of EOC is based on four main techniques: pelvic palpation examination (PPE), imaging (which includes transvaginal ultrasound or sonography, magnetic resonance imaging, computed tomography, and positron-emission computed tomography [22,23,24]), serum levels of specific proteins, and surgery (either laparoscopic or laparotomic) [25]; however, there is an urgent need to develop alternative techniques for early-stage HGSOC identification. Accordingly, new diagnostic methods based on cellular techniques or molecular approaches—such as gene expression profiling via NGS—are under development.

### 2.1. Molecular Markers and Algorithm Decisions for the Diagnosis of HGSOC: Carbohydrate Antigen 125 (CA125), Alone or in Combination with Other Imaging Techniques or Biomarkers

Initial studies found elevated levels of CA125—a transmembrane cell-surface protein encoded by the *MUC16* gene—in HGSOC compared to healthy ovarian tissue [26,27]. CA125 remains the most-used molecular marker; however, the use of CA125 suffers from several drawbacks. Elevated CA125 levels occur in only 23–50% of stage I-II cases, and CA125 cannot be detected in all advanced HGSOC cases [28,29,30,31,32]. Moreover, female patients who smoke or develop inflammatory processes, as well as those with physiological or benign conditions (e.g., menstruation, pregnancy, uterine fibroids), might display altered serum CA125 levels, thereby increasing false positive rates and HGSOC misdiagnoses [33,34,35,36,37,38,39,40,41,42,43,44]. This fact highlights the need to conduct further studies to elucidate the relationship between these variables and CA125 levels. Therefore, CA125 levels alone cannot discriminate early HGSOC cases with sufficient sensitivity and specificity. To solve this problem, clinical trials and triage algorithms have explored the power of combining CA125 with imaging techniques for the early diagnosis of HGSOC. Additionally, distinct predictive indices based on biomarkers and ultrasound have been developed to differentiate the nature of adnexal masses, which have the advantage of eliminating inter- and intra-observer variability, estimating the probability of mass malignancy, and increasing efficacy and efficiency [45].

The Risk of Malignancy Index (RMI) algorithm combines CA125 levels, ultrasound results, and menopausal status [46]; however, despite the proposal of several combinations of the formula [47], the RMI possesses lower sensitivity than transvaginal sonography (TVS) alone [32,48]. For this reason, clinical trials such as the Prostate, Lung, Colorectal, and Ovarian (PLCO) Cancer Screening Trial evaluated CA125 levels combined with TVS; however, the results failed to provide evidence of improvements in mortality rates or early-stage HGSOC detection rates [30,31,49,50].

Other strategies based on the Risk of Ovarian Cancer Algorithm (ROCA) [51] aimed to stratify risk according to CA125 levels; however, the use of ROCA in the Normal Risk Ovarian Screening Study (NROSS) [52] or the United Kingdom Collaborative Trial of Ovarian Cancer Screening (UKCTOCS) [53] clinical trials failed to suffice for the early detection of OC due to a high number of false positives and the associated failure to reduce HGSOC-associated mortality rates [14,54]. Conversely, a study by Srivastava et al. that combined CA125 with additional protein biomarkers suggested some clinical impact [20]. The glycoprotein HE4 (human epididymis 4) displays elevated levels in HGSOC and endometrioid EOCs, presumably at advanced tumor stages [55,56]. Meanwhile, other biomarkers examined in combination with CA125 and HE4 include mesothelin [57], CEA and VCAM-1 [58], glycodelin [59], IL-6 and E-cadherin [60], and transthyretin [61]. The Assessment of Different Neoplasias in the Adnexa (ADNEX) model represents an alternative strategy that considers six ultrasound and three clinical predictors (including CA125 levels) to discriminate between benign, borderline, invasive, and metastatic ovarian tumors [62].

Alternative strategies such as the Risk of Ovarian Malignancy Algorithm (ROMA), OVA1™, or OVERA™ deserve special consideration given their U.S. Food and Drug Administration (FDA) approval and their relevance to routine clinical practice. ROMA, which displays high performance in menopausal patients, evaluates HE4 and CA125 levels and stratifies patients with adnexal masses into low or high risk of malignancy [63,64]. HE4 levels are not usually modified by benign pathologies or external factors, and they display greater specificity in differentiating between malignant and benign tumors; furthermore, HE4 combined with CA125 outperforms the specificity and sensitivity in detecting cases missed when using CA125 alone [32,65,66,67,68]. OVA1™ comprises a multivariate index assay (MIA) test that examines five biomarkers (i.e., transthyretin, apolipoprotein A-1, 2-microglobulin, transferrin, and CA125) and scores the likelihood of malignancy of a pre-detected adnexal mass prior to surgical intervention [69,70,71]. OVERA™ examines levels of CA125, HE4, apolipoprotein A-1, FSH (follicle-stimulating hormone), and transferrin, achieving a sensitivity of 91% and specificity of 61% for HGSOC screening [72]. Other recently proposed protein panels have included CA125, vitamin-K-dependent protein Z, C-reactive protein, and LCAT [73]; CA125, HE4, CA72-4, and MMP-7 [74]; or CA125, HE4, FOLR1, KLK11, WISP1, MDK, CXCL13, MSLN, and ADAM8 [75]. The most recent strategies rely on the detection of autoantibody levels in combination with CA125 [76,77] and involve the detection of anti-TP53 [78], anti-HSF1, or anti-CCDC155 [79].

### 2.2. Gene Expression Profiling and Gene Panels

While ongoing gene expression profiling studies have suggested that HGSOC represents a highly heterogeneous pathology [6], expression analyses can differentiate between HGSCO and other types of EOC. For example, Sallum et al. reported a differential *WT1*, *TP53*, and *P16* expression profile that distinguishes HGSCO from LGSOC [80], while Li et al. found that 11 differentially expressed genes (DEGs) could discriminate between borderline cases and HGSOC tumors [81].

Different gene expression profiles reported by studies such as the Cancer Genome Atlas (TCGA) project [82,83] can be associated with pathological outcomes [84]. For instance, high expression of *HOX*, *FAP* (myofibroblast markers), and *ANGPTL1/2* (markers of microvascular pericytes) or high expression of *HMGA2*, *SOX11* (transcription factors), *MCM2*, and *PCNA* (proliferation markers) is correlated with worse prognosis [85,86,87]. Additionally, those studies linked improved survival rates and patient prognoses to *MUC* gene (*MUC16*) expression or *CXCL11*, *CXCL10* (chemokine ligands), and *CXCR3* (receptor) expression [85,86,87].

Gene expression panels such as *NR5A1*, *GATA4*, *FOXL2*, *TP53*, and *BMP7* possess different profiles when comparing primary OC tumors and their metastases, and could eventually be used to predict patient survival [88]. Additionally, the expression of homologous recombination repair (HRR) genes (e.g., *BRCA1*, *ATR*, *FANCD2*, *BRIP1*, *BARD1*, and *RAD51*) is associated with a better prognosis in HGSOC cases, whereas the expression of epithelial-to-mesenchymal genes (e.g., *GATA4*, *GATA6*, *FOXC2*, *KLF6*, and *TWIST2*) is associated with a worse prognosis [89].

Studies have also assessed differential gene expression in HGSOC to guide the optimal therapeutic choice and measure expected responses [84]. A meta-analysis by Matondo et al. that included 1020 patients identified a prognostic signature regulated by *HIF1α* and *TP53* in therapy-unresponsive patients as an indicator of a worse overall prognosis [90]. Lee et al. also identified several DEGs in patients who underwent complete gross resection or neoadjuvant chemotherapy with either positive or negative responses [91]. Most recently, Buttarelli et al. identified a 10-gene signature (including genes such as *CTNNBL1*, *CKB*, *GNG11*, *IGFBP7*, and *PLCG2*) for classifying wild-type BRCA HGSOC patients into sensitive- and resistant-to-therapy groups [92].

Novel bioinformatic approaches are currently attempting to refine gene expression signatures that predict therapeutic responses [93] and differentiate between HGSOC and other cancer types [94]. For example, co-expression network analyses have identified *UBE2Q1* as a prognostic biomarker associated with poor relapse-free overall survival in HGSOC patients [95].

Overall, using DEG analysis in early diagnosis has the potential to improve the management and survival of HGSOC patients.

## 3. Novel Trends in the Understanding of the Origin of HGSOC

### 3.1. Cell Origin

Scientists and physicians have attempted to elucidate the possible cellular origin of HGSOC for over a century; unfortunately, a clear origin has yet to be established, since no specific precancerous lesion has been recognized, which represents a significant obstacle to early diagnosis approaches [6,96,97].

The first theory for HGSOC’s cell of origin points to the ovarian surface epithelium (OSE) (Figure 1). Fathalla (1971) suggested a putative relationship between ovulation and the development of ovarian neoplasms [98], giving rise to other studies assessing this link [99,100,101]. While the pro-inflammatory and pro-oxidative environment generated in the OSE due to ovulation-induced tissue rupture may result in cell and DNA damage [102,103], the induction of so-called cortical inclusion cysts (CICs—sections of the OSE that invaginate and remain trapped beneath the ovarian surface during ovulation cycles [104]) is also regarded as a potential origin of HGSOC. 

The observation of genetically and phenotypically altered cells (i.e., protoneoplastic lesions) in the ciliated end of the fallopian tube in female patients carrying *BRCA1* or *BRCA2* mutations undergoing salpingectomies [105,106] provided support for the fallopian tube epithelium (FTE) as the principal site for the origin of HGSOC [107,108,109,110,111] (Figure 1). These lesions were later described as “serous tubular intra-epithelial carcinomas” (STICs) and might arise due to the influence of the pro-inflammatory and pro-oxidative factors released during ovulation [102,112,113]. STICs and HGSOC share several features [6,96], including the presence of *BRCA* gene mutations [114,115], identical *P53* gene mutations, and a strong correlation in *CCNE1* copy number amplification [116,117]. STICs also display a so-called p53 signature: secretory cells in the distal fallopian tube characterized by intense TP53 staining (proof of p53-mutated protein), positive γ-H2AX staining (a DNA damage marker), and the absence of Ki-67 staining (minimal proliferation activity) [116,118,119]. Indeed, a p53 signature may highlight precursor lesions to STICs, as they also display genomic instability (a genomic feature of HGSOC) [97] and telomere shortage (an early sign of HGSOC onset) [120,121]; therefore, the p53 signature may be necessary but not sufficient for the development of HGSOC [122,123]. Lately, a more complex theory has suggested that secretory epithelial cells from the distal fallopian tube (where STICs are found) can implant into the OSE, thereby resembling CICs of tubal epithelial origin [124], which would explain cases where HGSOC does not arise directly on the fallopian tube [13].

Although recent evidence suggests the likelihood of a tubular origin of HGSOC precursor lesions, a possible dual origin (OSE or FTE) for this tumor is also currently accepted [125,126,127,128]. Thus, future research efforts may focus on designing novel studies that establish biomarkers to distinguish between FTE and OSE origin as a potential preliminary step to achieve differential and early diagnoses.

### 3.2. Hormones

The heterogeneous expression of estrogen and the estrogen receptor (ER) alters depending on tumor stage and subcellular location and even changes with different prognoses in HGSOC patients [7,129,130]; however, hormones such as FSH, luteinizing hormone (LH), androgens, and their respective receptors are associated with disrupted cell proliferation and serous EOC development [129,130,131,132,133]. Conversely, studies have highlighted the protective nature of progesterone (P4) [134,135]. In this regard, two theories are considered:(a) Gonadotropin hypothesis: The risk of serous EOC increases due to the excessive ovarian tissue uptake of FSH and LH [136,137]. The FSH-mediated proliferation and migration of EOC cells via SphK [138] and the FSH-R/LH-R-mediated cell migration and invasiveness via COX2 [139,140] or ERBB-2 [141] support this theory.(b) Androgen/progestin hypothesis: This dual hypothesis acknowledges high androgen levels (usually linked to polycystic ovarian syndrome (PCOS) or obesity) as an EOC risk factor [142,143,144] and P4 as a protective factor [132,133]. High expression of AR [145,146] in HGSOC [147,148] and FTE is associated with the onset of serous EOC [149]. Conversely, PR expression is associated with a favorable prognosis and a reduced metastatic risk [134,150].

In line with these two hypotheses, differing physiological conditions could be considered to be risk factors (e.g., ovulation and menopause) or protective factors (e.g., pregnancy, breastfeeding, oral contraceptives). During ovulation and menopause, elevated FSH and LH flux from the pituitary gland to the ovarian epithelium increases the risk of tumor development [151]. Indeed, a recent meta-analysis concluded that a higher risk of serous EOC is associated with more lifetime ovulatory years [152], while another publication linked the administration of hormone replacement therapy (HRT) to increased risk of serous EOC during menopause [153]. The link between factors suppressing ovulation and the putative reduction in an individual′s risk of developing the disease has been evaluated by several authors [154]. Pregnancy could exert a protective effect against EOC due to the anovulation period, the strong negative regulation of FSH and LH secretion, and increased P4 secretion [151]. The association between parity and a trend toward a reduced risk of serous EOC further supports this protective role [155,156]. Such protection against EOC has also been attributed to factors such as breastfeeding and the consumption of oral contraceptive drugs [157]. Breastfeeding is associated with a reduction in the risk of OC, which may persist for several years depending on the number of breastfeeding episodes and the age at first feeding [158,159]. The protective role exerted by oral contraceptives has been found to increase with the length of time they are used [160,161].

Based on this knowledge, strategies that consider the expression of AR, ER, and PR have been explored to predict survival rates and responses to therapy in HGSOC [148,162].

### 3.3. Genetic Traits

The complex genomic landscape of HGSOC, which arises from a combination of inherited and/or somatic mutations and chromosomal abnormalities, epigenetic alterations, and signaling pathway dysregulation (Table 1), has been studied in the search for traits that predict patient prognosis [163] or response to therapy [164]. The location of the most frequent genetic alterations and an understanding of the contribution of each mutation to pathology may be significant; however, not all reported mutations affect HGSOC’s malignant transformation equally [165]. Thus, more in-depth knowledge regarding tumor mutational burden may contribute to defining biomarkers that support the early diagnosis of HGSOC.

#### 3.3.1. Inherited Mutations

HGSOC possesses a robust heritable component involving germline mutations that contribute to the development of disease in 25% of cases [183]. Germline mutations in the *BRCA1* and *BRCA2* genes, present in 13–22% of HGSOC cases [184,185,186], are tightly correlated with the onset of OC, given their detection in patients who developed HGSOC from precancerous FTE lesions [97]. Other reported germline mutations include *RAD51*, *BRIP1*, *PALB2*, *CHEK2*, *MRE11A*, *RAD50*, and *ATM*, which belong to the Fanconi anemia–BRCA pathway and play crucial roles in the HRR system for double-stranded DNA breaks [187,188,189,190]. Mutations affecting genes involved in HRR may also trigger HRR pathway deficiency (HRD), thereby increasing DNA errors, genomic instability, and the risk of HGSOC [82,191,192]. In this sense, HRD testing provides a new tool for the stratification of HGSOC that might simplify physicians’ decisions when managing and monitoring responses to therapy in affected patients [193,194]. New HRD testing approaches have assessed the performance of biomarkers such as *RAD51* [195] or calculated the levels of genomic instability based on sequence variants in the *BRCA* genes [196].

Genome-wide association studies (GWASs) have revealed associations between single-nucleotide polymorphisms (SNPs) and pathogenic phenotypes within the population [197] or the incidence and survival rates among ethnicities [198]. Studies have reported an additional percentage of estimated heritability for HGSOC from SNP characterization at susceptible loci (e.g., 3q28, 4q32.3, 8q21.11, 10q24.33, 18q11.2, and 22q12) [166,199].

As most SNPs occur in non-coding regulatory regions, transcriptome-wide association studies (TWASs) and expression quantitative trait locus (eQTL) analyses have sought to overcome this drawback when studying disease predisposition. Lawrenson et al. described three statistically significant eQTL associations for *HOXD9*, *CDC42*, and *CDA8* [167], while Gusev et al. reported an additional 23 candidate genes [168], indicating their putative role in early-stage HGSOC. A study combining breast and ovarian cancer GWAS datasets and transcriptomic data reported 11 candidate susceptibility genes, including *CCNE1*, *CPNE1*, *HEATR3*, and *STRCP1* [169]. Novel analyses integrating candidate genes and gene regulation have encountered additional genes linked to an increased risk of HGSOC [200]. These data emphasize the importance of genome elements in non-coding regions, even though the precise mechanisms underlying these associations remain unclear.

#### 3.3.2. Somatic Mutations

*TP53*, the most frequently mutated gene associated with HGSOC, functions as “the guardian of the genome” by regulating upstream or downstream genes to control tumor suppression and maintain cellular homeostasis [201,202]. The TCGA project and subsequent studies have reported *TP53* mutations in around 93–96% of HGSOC patients [82,172,201,203]. A study by Cheng et al. that compared early and late HGSOC mutational landscapes reported *TP53* mutation rates of 100% in late-stage cases and 82% in early-stage cases [170], consistent with previous results showing the very frequent but not global *TP53* mutational load in HGSOC patients [204]. *TP53* mutations are mostly missense (60.52%) but can also be frameshift (15.24%), splice-site (10.53%), nonsense (10.73%), and in-frame (3.22%) mutations [82]; however, a recent study by Park at al. reported even higher rates of missense mutations (62.5% and 95.8%, analyzed by NGS and immunohistochemistry (IHC), respectively) [205]. R273, R248, R175, and Y220 represent the most reported mutation hotspots in the DNA-binding domain of the TP53 sequence [205,206,207] and may define differential features between early and late stages of HGSOC [170,203,208,209,210,211,212]. HRR genes such as *BRCA1/2* and *RAD51D* may also undergo somatic mutations [183,213,214]. According to the TCGA project and succeeding studies, other mutated genes with a role in HGSOC include *CSMD3* and *FAT3* (tumor-suppressor genes), *MLH3* (with known roles in DNA mismatch repair), and *CDK12* (RNA splicing regulation) [82,91,171].

Corona et al. described several point mutations that converge on TEAD4/PAX8-binding sites during OC’s progression [215], highlighting the role of non-coding elements in the development of OC. Moreover, Ni et al. reported several mitochondrial genes with heteroplasmic somatic DNA mutations that may confer selective advantages to tumor cells [216]. These data may help to stratify different HGSOC cases—a promising strategy to consider when seeking early diagnostic tools [217].

#### 3.3.3. Chromosomal Aberrations

Chromosomal instability (CIN) leads to aberrant gene expression patterns and protein functions related to primary tumor growth/development and metastatic burden in several cancers [218]. Specifically, gains in chromosomes 1, 3, 7, 8, 12, and 20 and losses in chromosomes 5, 6, 11, 16, 17, 18, 19, and 22 characterize HGSOC [82,173]. Identifying such patterns may help to predict tumor behavior and define early diagnostic strategies (as reported by Drews et al. [219]). Chromosomal aberrations in HGSOC may be solely due to ubiquitous *TP53* mutations [220], cell-cycle imbalances promoted by mutations in HRR genes, or alterations in regulator proteins such as AURKA or CNNE1 [221]. CIN affects genes located in focal amplification areas (e.g., *CNNE1*, *MYC*, and *MECOM*) and focal deletion areas (e.g., *PTEN*, *RB1*, and *NF1*) [82,172]; consequently, patients harboring these aberrations usually evade therapy and display tumor recurrence, as occurs for *CNNE1* amplification [222] or *PTEN* loss [223]. Interestingly, focal amplifications, deletions, and copy number signatures display different profiles in early-stage and late-stage HGSOC, thereby representing a promising option when seeking strategies for early diagnosis [170].

Chromothripsis, a well-reported form of CIN with a critical role in cancer [224], is correlated with poor responses to therapy in HGSOC [91]. Interestingly, Engqvist et al. reported 46 chromothripsis-like patterns (CLPs) for HGSOC stages I and II, which affected cancer-related genes such as *MLF1*, *BRCA1*, *CCNE1*, *TP53*, *ARID1A*, *MYC*, and *PIK3CA* [174]. Finally, whole-genome duplication (WGD), which leads to CIN, is frequently encountered in late-stage HGSOC due to a relationship with *TP53* mutations, *CNNE1* amplification, or *RB1* loss [170,225].

### 3.4. Epigenetics

Epigenetic mechanisms involving DNA methylation (5-methylcytosine; 5 mC), histone modifications, or non-coding RNAs (ncRNAs) can regulate gene expression, participate in the onset of HGSOC, and influence therapeutic outcomes [4,226,227].

DNA methylation is generally associated with decreased gene expression and can affect the promoters of tumor-suppressor genes. For instance, *BRCA1* promoter hypermethylation in families affected by HGSOC without BRCA1/2 germline mutations is correlated with allelic *BRCA1* loss [175]. Hypomethylation also generates genomic instability through the enhanced transcription of oncogenes [176,177]. Moreover, decreased levels of 5-hydroxymethylcytosine (5-hmC—an oxidized form of 5 mC) represent a hallmark indicator of malignancy and tumor progression in HGSOC patients [228].

Histone modifications carried out by methyltransferases (HMTs) or deacetylases (HDACs) have also been implicated in the development of HGSOC [229]. Specifically, overexpression of the CARM1 protein arginine methyltransferase (PRMT) occurs in breast cancer and HGSOC cases [230], leading to the transcriptional repression of tumor-suppressor genes such as *NOXA* [178]. Furthermore, overexpression of HDAC in advanced OC stages (i.e., III and IV) compared to early stages (i.e., I and II) may represent a predictive factor and a target to prevent OC progression [179,231].

The expression of ncRNAs—specifically, microRNAs (miRNAs) and long non-coding RNAs (lncRNAs)—represents an additional epigenetic mechanism with importance in the pathophysiology of cancer [4,232]. A study including 894 EOC cases found 16 and 19 miRNAs to be associated with better and worse prognoses, respectively [233]. Overexpression of miR-1290 occurs in HGSOC patients compared to healthy controls [180], while miR-27-a-3p expression in HGSOC patients is correlated with a worse prognosis [181]. Other ncRNAs—such as miR23a, the miR200 family, and miR205—participate in cell proliferation, migration, apoptosis, invasion, or metastasis in OC cases [182,234]. Finally, Wang et al. observed the differential expression of 633 lncRNAs in malignant EOC compared with benign and normal conditions [235]. In a specific example, Liu et al. described the upregulation of the *CTBP1*-*DT* lncRNA and its link to the malignant behavior of HGSOC cells and disease progression, suggesting this ncRNA as a possible biomarker for early diagnosis or a therapeutic target [236].

### 3.5. Target Signaling Pathways in HGSOC

Relevant signaling pathways involved in HGSOC include RB (67% of cases), PI3K/RAS (45% of cases), NOTCH (22% of cases), and FOXM1 (85% of cases) [82,172,237] (Figure 2).

The RB pathway controls the G1-to-S phase transition in mammalian cells [238,239]. Overactivation of the RB pathway prompts the increased activity of the E2F transcription factor compared to normal tissues, leading to exacerbated cell proliferation due to an uncontrolled cell cycle [240]. Additionally, genomic conditions involving *RB1* loss combined with HRD are associated with unusually long survival rates after chemotherapy, providing opportunities for more precise stratification of HGSOC [241].

Alterations in the PI3K/Akt/mTOR pathway in HGSOC mainly derive from genomic amplification of *PIK3CA* (20% of cases) or *AKT* isoforms (15–20% of cases) [242,243]. None of the tested inhibitors of this pathway are currently used in clinical practice, due to the lack of successful results in previous studies [244,245]. Ras/Raf/MEK pathway alterations observed in HGSOC cases derive from copy number changes such as *KRAS* oncogene amplifications (11% of cases), *MAPK* amplification (20% of cases), *NF1* loss (8% of cases), and other less frequent changes in *NRAS* or *BRAF* [82].

Interestingly, *BRCA1/2* promoters represent downstream targets of the RB, PI3K, and RAS signaling pathways, offering opportunities for therapeutic interventions using CDK, RTK, and MAPK inhibitors such as cediranib, nindetanib or pazopanib (which unfortunately have not demonstrated efficacy in terms of improving disease-free interval or survival [246,247]), or PARP inhibitors such as niraparib (first-line treatment for these tumors) or olaparib and rucaparib (specifically targeting *BRCA*-positive tumors) [239,248]. Other drugs currently used for treating HGSOC include bevacizumab, directed against the vascular endothelial growth factor (VEGF) [249,250].

Different molecular traits, such as *E2F* copy number gain and amplification or the overexpression/inactivation of *FOXM1* upstream transcription factors such as ERBB2 (tyrosine kinase receptor) or TP53, might cause overexpression of *FOXM1* in HGSOC. Abnormal *FOXM1* overactivation, in addition to copy number gains on this gene, coincides with abnormal cell-cycle processes that promote cell proliferation, cancer stemness, genomic instability, and poor prognosis [251,252].

Notch family members are transmembrane receptors that, upon binding and cleavage, migrate to the nucleus and target the promoters of genes such as *CCND*, *p21CIP1*, *NF-κβ*, and c-*MYC* [237]. Other factors related to Notch signaling disruption include *CXCR4/SDF1α* signaling dysregulation [253] or *DK1* overexpression (non-canonical Notch ligand) [254], with both being related to tumor progression, higher epithelial-to-mesenchymal transition, and worse prognosis in HGSOC.

## 4. Ongoing Clinical Trials

The upward trend in the number of clinical trials focused on OC—and on HGSOC in particular—has translated into an increase in survival and positive outcomes when comparing the last five years with the early 2000s [255].

Up to 949 clinical trials have been registered in the clinicaltrials.gov portal of the U.S. National Institutes of Health (NIH) within the last five years [256], with 19 related to early diagnosis. Most clinical trials involve the detection of different genetic traits in Pap smears and/or uterine lavage or blood (so-called “liquid biopsies”) that might guide the diagnosis of HGSOC. The latest advances include the detection of differential DNA methylation levels by NGS (NCT0362238, NCT04651946), tumor-educated platelets (TEPs) and circulating tumor DNA (ctDNA) (NCT04022863, NCT04971421), biomolecules via plasma spectroscopy (NCT04817449), blood proteins (NCT04794322), cell-free DNA (cfDNA) (NCT04261972, NCT04511988), circulating (ct)RNAs (e.g., miRNAs and lncRNAs) (NCT03738319), and other tumor-associated changes in circulating glycoproteins (NCT03837327).

It should be noted that while much existing research has focused on early and non-invasive detection, the need for a very large study population for general screening represents an obstacle to the design of such studies [257,258,259].

## 5. Liquid Biopsy as a Non-Invasive Tool for the Early Diagnosis of HGSOC

Liquid biopsies, based on the non-invasive detection of molecular biomarkers (mainly through NGS) released from tumor cells into the bloodstream, represent an alternative approach to the early detection of HGSOC [260]. Here, we review current evidence regarding the detection of circulating tumor cells (CTCs), ctDNA, ctRNA, and extracellular vesicles (EVs) [261] (Figure 3).

### 5.1. Circulating Tumor Cells

CTCs are released from a primary or metastatic tumor and shed into the peripheral blood, where they can be isolated by size selection [262] or by targeting surface proteins such as EpCAM, MUC1, or HER2 [263]. Benign masses release lower CTC counts than EOC tumors [264,265], as corroborated by studies reporting that patients with stage I-II and stage III-IV are 8.4 and 16.9 times more likely, respectively, to harbor CTCs than patients with benign adnexal masses [266,267]. Recent bioinformatic approaches have explored the International Federation of Gynecology and Obstetrics (FIGO) stages of preoperative EOC patients based on CTCs [268]. Although CTCs possess a greater predictive value than CA125 for early tumor stages (IA-IB) [263], they display low survival rates in blood circulation, making them more suitable for monitoring disease progression or metastasis [269,270] than for early detection [271].

### 5.2. Cell-Free DNA and Circulating DNA

ctDNA is the fraction of total cfDNA released from tumor cells, mainly through necrosis and/or apoptosis [272]. ctDNA levels in healthy patients or patients with benign conditions remain low [273,274,275] but increase proportionally according to tumor burden and stage (i.e., higher ctDNA levels when advancing from stage III to IV), reaching maximum levels when metastasis develops [276,277,278].

Early detection strategies using ctDNA rely on evaluating methylation patterns and detecting somatic mutations or aberrant fragmentation patterns [260,279,280]. Promoter hypermethylation patterns in genes such as *OPCML*, *RASSF1A*, *RUNX3*, and *APC* [281,282,283], *COL23A1*, *C2CD4D*, and *WNT6* [284], or *HOXA9* and *HIC1* [285] have been detected in ctDNA from early-stage HGSOC cases. *TP53* mutations and other somatic variants have also been detected in ctDNA from stage I and II HGSOC cases [286,287,288]. ctDNA analyses in combination with other markers also deserve consideration [289,290], as evidenced in a study by Cohen et al. that described a combination of a 61-amplicon panel with serum protein biomarkers for the early diagnosis of EOC [291]. Finally, ctDNA fragmentation patterns, which display irregularity in cancer patients compared to healthy individuals, are currently regarded as a novel strategy for predicting the risk of OC [280,292].

### 5.3. Cell-Free RNA and Circulating RNA

Detection of ctRNA, as a fraction of the total cfRNA released from tumor cells, represents a promising approach in liquid biopsies, since this strategy offers a snapshot of the transcriptional landscape and mirrors potential pathogenic processes [293]. ctRNA has mostly been reported as a prognostic maker but has been explored in detecting lung or breast cancers [294,295], suggesting that early diagnosis of HGSOC could take advantage of ctRNA levels. Currently, the few diagnostic markers contemplated for early diagnosis are miRNAs (e.g., *miR-1246*, *miR-595*, and *miR-2278*) [296,297] overexpressed in HGSOC tumors [298] or lncRNAs (e.g., MIR4435-2HG [299] and *CASC11* [300]) upregulated in OC. It should be noted that further research will need to determine the diagnostic sensitivity and specificity of ctRNA for early tumor detection and standardize protocols for isolation and analysis [301].

### 5.4. Exosomes

Exosomes are a subtype of EVs that are released from tumors and extravasate into the bloodstream, where they can avoid the immune system [302], promote metastasis in sites distant from the primary tumor [260,303], and promote the development of chemotherapy resistance [304]. Several components of the exosomal “cargo” can be isolated when seeking tumor-specific features [305]. A panel of eight miRNAs (miR-141, miR-21, miR-200a, miR-200b, miR-200c, miR-214, miR-205, and miR-203) [306], miR-200c, miR-93, and miR-145 [307], and miR-34-a [308] have shown potential as diagnostic biomarkers for early OC detection. Furthermore, proteins such as LBP, GSN, GFA, and FGG have also been considered as potential biomarkers [309]. The latest research evaluating exosomal proteomes from FIGO stage I and II HGSOC cases confirms the potential of proteins enclosed in exosomes in early-stage OC screening [310].

## 6. Conclusions

Ranked as one of the most aggressive and deadly forms of gynecological cancer, HGSOC is currently considered to be a public healthcare issue that significantly impacts female patients’ quality of life. Current challenges arise from the difficulty in establishing an early and effective diagnosis in patients with adnexal masses. HGSOC is mostly diagnosed in advanced FIGO stages (III-IV) due to the lack of symptoms and the use of traditional diagnostic methods. The performance of CA125—the only available molecular serum biomarker—alone or in combination with other biomarkers or TVS, has yet to support significant reductions in mortality rates.

Other strategies have focused on the possible dual origin of HGSOC (the FTE or OSE) and the influence of hormones in promoting or protecting against the development of disease. Furthermore, HGSOC is characterized by extensive genomic instability, promoted by almost universal mutations in the *TP53* gene, genes belonging to the HRR system, and CIN. Furthermore, alterations in multiple signaling pathways and epigenomic mechanisms have also been described.

Research in this field is rapidly moving forward, allowing the molecular management of HGSOC by encompassing multiple disease-associated features in high-throughput, personalized approaches. Proof of such advances includes the putative gene panels and expression analyses intended for the stratification of HGSOC, the establishment of treatment-guided decisions, and the monitoring of disease progression, which can be extended to early detection.

Most recently, liquid biopsies have gained momentum as an emerging strategy in the field of gynecological oncology, providing a non-invasive approach with promising applications from several clinical standpoints, including early diagnosis of adnexal masses or monitoring of HGSOC patients’ prognosis and treatment effectiveness.

This review sought to provide an update of those studies concerning the molecular and cellular characterization of HGSOC tumors, which may serve as a starting point for the design of new strategies aimed at improving management of this subtype of cancer.

## Figures and Tables

**Figure 1 ijms-23-13777-f001:**
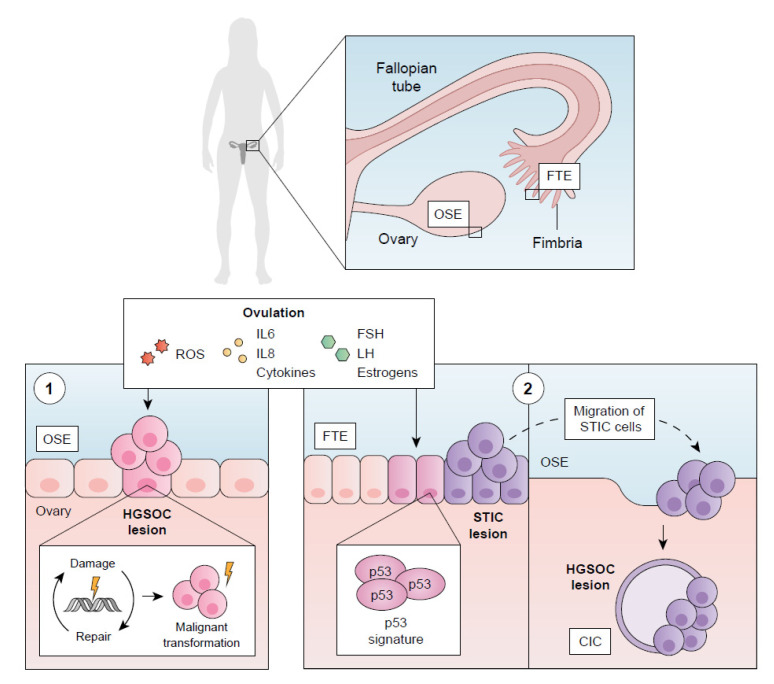
Two main theories of the cellular origin of high-grade serous ovarian carcinoma (HGSOC): (**1**) Reactive oxygen species (ROS), inflammatory molecules (IL-6, IL-8, and cytokines), and hormones (FSH, LH, and estrogens) released during ovulation induce constant tissue damage and repair cycles, leading to HGSOC in the ovarian epithelial surface (OSE), which might also be triggered by the formation of cortical inclusion cysts (CICs). (**2**) Alternatively, malignant transformation triggered by the pro-inflammatory and pro-oxidative environment generates serous tubular intra-epithelial carcinoma (STIC) lesions in the fallopian tube epithelium (FTE), which give rise to HGSOC after migrating and invaginating in the OSE as CICs.

**Figure 2 ijms-23-13777-f002:**
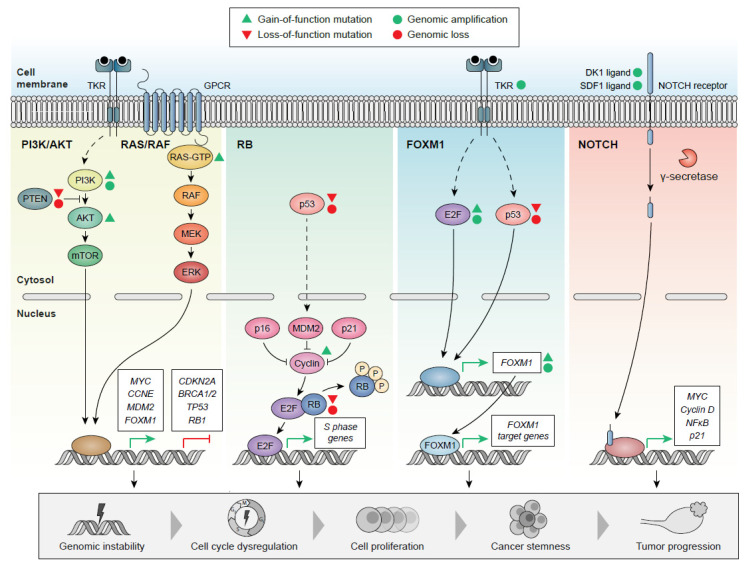
Frequently altered signaling pathways in HGSOC: Different mutations and genomic aberrations affect the PI3K/Akt, Ras/Raf, Rb, FOXM1, and Notch pathways, resulting in the dysregulation of downstream target gene expression, which can subsequently promote genomic instability, cell-cycle dysregulation, enhanced cell proliferation, cancer stemness, and tumor progression.

**Figure 3 ijms-23-13777-f003:**
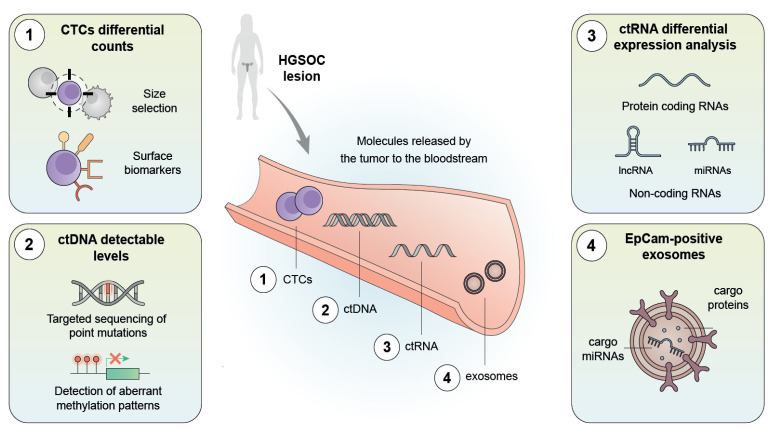
Liquid biopsy strategies for the early detection of HGSOC: Liquid biopsies rely on isolating CTCs, ctDNA, ctRNAs, or exosomes released by tumors that enter the bloodstream. Different analytical techniques support the identification of putative biomarkers for the early diagnosis of HGSOC.

**Table 1 ijms-23-13777-t001:** Genetic and epigenetic traits involved in the etiopathogenesis of HGSOC. Major genetic alterations include inherited mutations, somatic mutations, chromosomal aberrations, and epigenetic alterations, which may be considered for the early diagnosis, prognosis, and follow-up evaluation of HGSOC.

**Inherited Mutations**	**Mutations**		**Susceptible Loci**		**eQTL Associations**		[82,166,167,168,169]
	*BRCA1/2*, *CHEK2*, *RAD51*, *MRE11A*, *BRIP1*, *RAD50*, *PALB2*, *ATM*		3q28, 10q24.33, 4q32.3, 18q11.2, 8q21.11, 22q12		*HOXD9*, *STRCP1*, *CDC42*, *CPNE1*, *CDA8*, *HEATR3*, *CCNE1*		
**Somatic Mutations**	**Universal Mutation**			**Other Mutations**			[82,91,170,171]
	*TP53*, in almost all cases			*PTEN*, *KRAS*, *FAT3*, *CSDM*, *MLH3*, *RB1*			
**Chromosomal Aberrations**	**Chromosomal Gains**	**Chromosomal Losses**	**Focal Amplification Areas**	**Focal Deletion Areas**	**CLP**		[82,172,173,174]
	Chr 1, 3, 7, 8, 12, 20	Chr 5, 6, 11, 16, 17, 18, 19	*CNNE1*, *MYC*, *MECOM*	*PTEN*, *RB1*, *NF1*	*MLF1*, *ARID1A*, *BRCA1*, *MYC*, *CCNE1*, *PIK3CA*, *TP53*		
**Epigenetic Alterations**	**DNA methylation**		**Histone Modifications**		**miRNAs**	**lncRNAs**	[175,176,177,178,179,180,181,182]
	Hypermethylation: ↓ transcription BRCA1 promoter		Methyltransferase overexpression (*CARM1*)		*miR-1290*, *miR27-a-3*, *miR23a*, *miR200 family*, *miR025*	*CTBP1-DT*	
	Hypomethylation: ↑ oncogene transcription		Histone deacetylase overexpression				

Chr: chromosome; CLP: chromothripsis-like patterns; eQTL: expression quantitative trait locus. (↑) indicates increase in transcription and (↓) indicates decrease in transcription.

## Data Availability

Not applicable.

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
