# Peer review of "Molecular Management of High-Grade Serous Ovarian Carcinoma"

_ijms, 2022, doi:10.3390/ijms232213777_

Round 1
Reviewer 1 Report
Dear author
Thank you for the submission of your review paper to our journal. In breast cancer that develops in superficial organs, many early-stage cases have been discovered with the introduction of mammography screening, leading to a decrease in the mortality rate of breast cancer. On the other hand, for ovarian cancer that develops deep in the pelvis, there is no established method for early detection, and early detection is difficult even in today's medical science, and the prognosis of ovarian cancer patients is still poor. This paper also mentions the genesis of BB and indicates the future direction of liquid biopsy, etc., and is expected to be useful for researchers in this field except for some points mentioned below.
Line 51
What’s the meaning of “histological premises”?
Line 75
The sentence “Moreover, female patients who smoke or develop inflammatory processes, as well as those with physiological or benign conditions (e.g., menstruation, pregnancy, uterine fibroids) also display elevated serum CA125 levels, thereby increasing false positive rates and HGSOC misdiag- 78 noses [30–37]. “ is not always true.
A study reported that mean CA 125 levels were lower in current smokers. Gynecol Oncol. 2008;110(3):383.
Line 271
The expression "Genome-wide association studies (GWAS) studies” is strange.
Author Response
Dear author
Thank you for the submission of your review paper to our journal. In breast cancer that develops in superficial organs, many early-stage cases have been discovered with the introduction of mammography screening, leading to a decrease in the mortality rate of breast cancer. On the other hand, for ovarian cancer that develops deep in the pelvis, there is no established method for early detection, and early detection is difficult even in today's medical science, and the prognosis of ovarian cancer patients is still poor. This paper also mentions the genesis of BB and indicates the future direction of liquid biopsy, etc., and is expected to be useful for researchers in this field except for some points mentioned below.
We thank the reviewer for their opinion and positive feedback, as well as, for taking the time to review our manuscript.
Point 1: Line 51. What’s the meaning of “histological premises”?
Response 1: Authors appreciate this comment since it might result confusing in some way. From our point of view, “histological premises” refers to those histology-based techniques that are currently employed for diagnostic confirmation in patients with adnexal masses. In this regard, biopsies of the suspected tumor are usually taken, being subsequently evaluated by an anatomic pathologist to finally confirm the diagnoses of HGSOC or any other ovarian alteration.
Per your request, authors have replaced the term "histological premises" to "histological evaluation" in case it might be clarifying and less confusing than the previous one.
In the text, these changes can be found in line 51.
Point 2: Line 75. The sentence “Moreover, female patients who smoke or develop inflammatory processes, as well as those with physiological or benign conditions (e.g., menstruation, pregnancy, uterine fibroids) also display elevated serum CA125 levels, thereby increasing false positive rates and HGSOC misdiagnoses [30–37]. “is not always true. A study reported that mean CA 125 levels were lower in current smokers. Gynecol Oncol. 2008;110(3):383.
Response 2: Thank you for this suggestion. After considering your comment we realized that there is some controversy on CA125 levels in smokers. As a result, we have included references supporting the decrease of CA125 levels in smoker patients (Pauler et al., 2001; Johnson et al., 2008; Fortner et al., 2017; Sasamoto et al., 2019), as well as the increase of the same biomarker in female patients who smoke that we already reported (Lycke et al., 2018). We have also clarified that more studies are needed to demonstrate the relationship between exposure to tobacco and the levels of this biomarker.
In the text, the changes can be found from line 80 to line 83.
Point 3: Line 271. The expression "Genome-wide association studies (GWAS) studies” is strange.
Response 3: We apologize for the mistake in typing. We have removed the word "studies", that is repeated (line 287).
Reviewer 2 Report
In this review, authors provide a detailed summary of the current status of nanoparticles for the delivery of antiviral, antifungal, and antiparasitic drugs into the body and the issues needed to be solved in the future.
Minor comment: in Table 2, in several places of C. albicans, periods are missing.
Author Response
Point 1: In this review, authors provide a detailed summary of the current status of nanoparticles for the delivery of antiviral, antifungal, and antiparasitic drugs into the body and the issues needed to be solved in the future.
Minor comment: in Table 2, in several places of C. albicans, periods are missing.
Response 1: Authors are grateful for the reviewer's comment, but unfortunately, we couldn’t address these questions since we think that the issues related to antiviral, antifungal, and antiparasitic drugs are far from our area of interest on ovarian cancer and we think that this comment belongs to another paper submitted to IJMS by other authors.
Authors will await the response of Reviewer #2 and the Editorial Board of IJMS.
Reviewer 3 Report
Original article about an extremely debatable subject that can be found in many works, which proves the interest of researchers.
I think some information should be well specified by the authors.
First, etiopathogenically, patients harboring germline mutations in BRCA1/2—which encode proteins involved in the pathway responsible for homologous recombination-mediated repair of double-strand breaks (DSBs)—have a much higher risk of developing HGSOC. According to this theory, the total number of ovulatory cycles a woman experiences would be directly related to her risk of acquiring HGSOC. Presence of small HGSOC-like dysplastic lesions in fallopian tubes of suspected BRCA mutation carriers. The existence of these microscopic intraepithelial carcinomas suggested that distal fallopian tube secretory epithelial cells (FTSEC) were the preferred cells of origin for HGSOC, at least in women carrying BRCA1/2 mutations. Despite compelling evidence that the fallopian tubes are the primary site of origin for HGSOC, it remains established that ovulation is a consistent risk factor in epidemiologic studies. Numerous studies have concluded that factors that suppress ovulation, such as pregnancy, breastfeeding and the use of drugs containing hormones. oral contraceptives have reduced an individual's risk of developing the disease.
Secondly, the limits of classical methods (clinical and imaging - attention is missing from the enumeration of CT, PET-CT techniques) and the need to develop alternative, cellular, molecular techniques for identification, especially in early stages.
Thirdly, it must be unequivocally emphasized that HGSOC represents a type of epithelial cancer, practically the only one, in which no precancerous lesion was recognized.
Fourthly, somewhat more extensively presented the role of molecular biology in therapy although the molecular biology of HGSOC does not present many oncogenic changes that can be easily targeted with small molecular inhibitors (eg PARP). A more recent example, the use of two PARP inhibitors, rucaparib and niraparib, to treat patients with relapsed ovarian cancer regardless of BRCA mutation status or platinum sensitivity. Other anti-angiogenic therapies aim to inhibit the VEGF receptor and other related receptor tyrosine kinases (RTKs) or another potential targeted therapy is the inhibition of AKT signaling.
Author Response
Comments and suggestions for authors: Original article about an extremely debatable subject that can be found in many works, which proves the interest of researchers.
I think some information should be well specified by the authors.
We are very grateful to the reviewer for his/her encouraging comment to improve the quality of our manuscript.
Point 1: First, etiopathogenically, patients harboring germline mutations in BRCA1/2—which encode proteins involved in the pathway responsible for homologous recombination-mediated repair of double-strand breaks (DSBs)—have a much higher risk of developing HGSOC. According to this theory, the total number of ovulatory cycles a woman experiences would be directly related to her risk of acquiring HGSOC. Presence of small HGSOC-like dysplastic lesions in fallopian tubes of suspected BRCA mutation carriers. The existence of these microscopic intraepithelial carcinomas suggested that distal fallopian tube secretory epithelial cells (FTSEC) were the preferred cells of origin for HGSOC, at least in women carrying BRCA1/2 mutations.
Despite compelling evidence that the fallopian tubes are the primary site of origin for HGSOC, it remains established that ovulation is a consistent risk factor in epidemiologic studies. Numerous studies have concluded that factors that suppress ovulation, such as pregnancy, breastfeeding and the use of drugs containing hormones oral contraceptives have reduced an individual's risk of developing the disease.
Response 1:
The authors agree with the reviewer's comment about including breastfeeding or oral contraceptives as protective factors of HGSOC tumorigenesis. As a result, we have also included references supporting the protective effect of breastfeeding (Modugno et al., 2019; Babic et al., 2020) or oral contraceptives (Havrilesky et al., 2013; Michels et al., 2018) in reducing individual's risk of developing HGSOC (Kotsopoulos et al., 2015; Trabert et al., 2020).
In the text, these data has been added from lines 241 to 257.
Point 2: Secondly, the limits of classical methods (clinical and imaging - attention is missing from the enumeration of CT, PET-CT techniques) and the need to develop alternative, cellular, molecular techniques for identification, especially in early stages.
Response 2: Authors apologize for missing CT and PET-CT when referring to imaging techniques. According to your suggestions, we have included them, also supported by their corresponding references (Jung et al., 2002; Sokalska et al., 2009; Shinagare et al., 2022).
Additionally, authors have remarked that there is a current gap in strategies for early diagnosis that should drive the development of alternative approaches.
Changes can be found in from lines 65 to 69.
Point 3: Thirdly, it must be unequivocally emphasized that HGSOC represents a type of epithelial cancer, practically the only one, in which no precancerous lesion was recognized.
Response 3: Authors agree with reviewer #3 about emphasizing the fact of no precancerous lesion has been identified previously in HGSOC.
Based on these premises, we have remarked this statement in lines 173 -174, also adding a new reference that supports this data (Labidy-Galy et al., 2017).
Point 4: Fourthly, somewhat more extensively presented the role of molecular biology in therapy although the molecular biology of HGSOC does not present many oncogenic changes that can be easily targeted with small molecular inhibitors (eg PARP). A more recent example, the use of two PARP inhibitors, rucaparib and niraparib, to treat patients with relapsed ovarian cancer regardless of BRCA mutation status or platinum sensitivity. Other anti-angiogenic therapies aim to inhibit the VEGF receptor and other related receptor tyrosine kinases (RTKs) or another potential targeted therapy is the inhibition of AKT signaling.
Response 4: Despite we have already mentioned the role of PARP inhibitors such as Niraparib, Rucaparib and, also Olaparib (Knudsen et al., 2010; Wiedemeyer et al., 2014), we have included other drugs targeting altered pathways in HGSOC that were missing in our review. Specifically, those related with RTK MAPK inhibitors such as Cediranib, Nindetanib or Pazopanib (Klempner et al., 2013; Rendell et al., 2022) or VEGF such as Bevacizumab (Burger et al., 2011; Nakai et al., 2022). Unfortunately, we could not report AKT inhibitors that are currently used in clinical activity, due to the lack of successful results in previous studies (Huang et al., 2020; van der Ploeg et al., 2021).
Changes can be found in from lines 400 to 413.